# Efficacy of Sildenafil in Patients with Severe COVID-19 and Pulmonary Arterial Hypertension

**DOI:** 10.3390/v15051157

**Published:** 2023-05-11

**Authors:** Oleksandr Valentynovych Oliynyk, Marta Rorat, Olena Vadymivna Strepetova, Serhij Oleksandrovych Dubrov, Vitaliy Grygorovych Guryanov, Yanina Volodymyrivna Oliynyk, Oleksii Serhijovych Kulivets, Anna Ślifirczyk, Wojciech Barg

**Affiliations:** 1Department of Anaesthesiology and Intensive Care, Bogomolets National Medical University, 01601 Kyiv, Ukraine; alexanderoliynyk8@gmail.com (O.V.O.); elestrepetova@gmail.com (O.V.S.); sergii.dubrov@gmail.com (S.O.D.); 2Department of Emergency Medicine, Rzeszow University, 35-310 Rzeszow, Poland; 3Department of Forensic Medicine, Wroclaw Medical University, 50-367 Wroclaw, Poland; 4Commercial Hospital “Manufaktura”, 08173 Kyiv, Ukraine; 5Commercial Hospital “Raiering”, 02121 Kyiv, Ukraine; joliynyk2@gmail.com (Y.V.O.); alex.kulivetz97@gmail.com (O.S.K.); 6Department of Medical Statistics, Bogomolets National Medical University, 01601 Kyiv, Ukraine; i_@ukr.net; 7Nursing Department, Siedlce University of Natural Sciences and Humanities, 08-110 Siedlce, Poland; aslifirczyk1@gmail.com; 8Department of Human Physiology, Rzeszow University, 35-310 Rzeszow, Poland; wbarg@ur.edu.pl

**Keywords:** SARS-CoV-2, respiratory failure, pulmonary artery pressure, sildenafil

## Abstract

Pulmonary arterial hypertension (PAH) is common in severe coronavirus disease 2019 (COVID-19) and worsens the prognosis. Sildenafil, a phosphodiesterase-5 inhibitor, is approved for PAH treatment but little is known about its efficacy in cases of severe COVID-19 with PAH. This study aimed to investigate the clinical efficacy of sildenafil in patients with severe COVID-19 and PAH. Intensive care unit (ICU) patients were randomly assigned to receive sildenafil or a placebo, with 75 participants in each group. Sildenafil was administered orally at 0.25 mg/kg t.i.d. for one week in a placebo-controlled, double-blind manner as an add-on therapy alongside the patient’s routine treatment. The primary endpoint was one-week mortality, and the secondary endpoints were the one-week intubation rate and duration of ICU stay. The mortality rate was 4% vs. 13.3% (*p* = 0.078), the intubation rate was 8% and 18.7% (*p* = 0.09), and the length of ICU stay was 15 vs. 19 days (*p* < 0.001) for the sildenafil and placebo groups, respectively. If adjusted for PAH, sildenafil treatment significantly reduced mortality and intubation risks: OR = 0.21 (95% CI: 0.05–0.89) and OR = 0.26 (95% CI: 0.08–0.86), respectively. Sildenafil demonstrated some clinical efficacy in patients with severe COVID-19 and PAH and should be considered as an add-on therapy in these patients.

## 1. Introduction

Severe acute respiratory syndrome coronavirus 2 (SARS-CoV-2) infection promotes the development of pulmonary arterial hypertension (PAH) [1]. This includes increased endothelial cell apoptosis induced by the virus entering the cells via the angiotensin-converting enzyme 2 (ACE-2) receptor, with subsequent microvascular corrosion and loss of vessel structure in the alveolar plexus, pericyte detachment from pulmonary microvessels with increased permeability, impaired smooth muscle cell differentiation, and proliferation with increased medial wall thickness, as well as increased fibroblast proliferation with fibrin and extracellular matrix deposition, resulting in adventitial thickening followed by parenchymal lung fibrosis [2]. Except for the direct causal factor, the pathomechanisms of COVID-19-induced pulmonary vasculopathy are similar to those in pulmonary arterial hypertension [3]. Most of the studies define PAH as systolic pulmonary artery pressure (sPAP) > 35–40 mm Hg [4,5]. The incidence of PAH varies depending on COVID-19 severity and is reported in 2% of patients with mild to moderate disease and nearly 50% of critical COVID-19 patients [6,7]. A meta-analysis by Oktaviono et al. [4] comprising 1728 COVID-19 patients from 16 studies demonstrated a PAH prevalence of 22% with a substantial impact on increased mortality (odds ratio, OR = 5.42), COVID-19 severity (OR = 5.74), and ICU admission (OR = 12.83), all *p* < 0.001. More recent studies present similar data [8,9].

Sildenafil is a potent and selective inhibitor of phosphodiesterase-5 (PDE-5), the enzyme that enhances smooth muscle relaxation via the nitric oxide/cyclic guanosine monophosphate (NO/cGMP) pathway [10]. Sildenafil had long been primarily used for the treatment of erectile dysfunction (Viagra) but PDE-5 is also abundant in the vascular, tracheal, and visceral smooth muscles. Thus, it started to be used for the treatment of PAH [11] and was approved for this purpose by the United States Food and Drug Administration (FDA) and the European Medicines Agency in 2005 and 2009, respectively. Consequently, sildenafil and other PDE-5 inhibitors are currently recommended for the treatment of PAH [12,13,14]. Therefore, the use of sildenafil in the treatment of PAH in the course of severe COVID-19 seems theoretically well justified [15,16,17,18]. Very little is known about the clinical efficacy of sildenafil treatment in patients with severe COVID-19 and PAH [19,20,21].

This study aimed to investigate the effect of sildenafil on the course and outcome of severe COVID-19 with PAH.

## 2. Materials and Methods

### 2.1. Study Setting

A randomized, double-blind, placebo-controlled, prospective clinical trial was carried out between 9 June 2020 and 12 February 2022 in the intensive care departments of three hospitals in Kyiv, Ukraine: City Clinical Hospital NO. 4, and two multi-profile commercial hospitals, “Manufaktura” and “Raiering”. The study was conducted on random Kyiv City inhabitants (caucasian). Patients with severe COVID-19 and pulmonary hypertension were randomly assigned to receive either sildenafil or a placebo. The primary endpoint was the mortality rate; secondary endpoints were the intubation rate and the duration of ICU stay.

### 2.2. Study Population

Adult patients with severe COVID-19 and PAH who met the following inclusion and exclusion criteria were recruited to the study.

Inclusion criteria:SARS-CoV-2 infection (confirmed with a positive reverse transcription polymerase chain reaction (RT PCR test));Bilateral interstitial pneumonia confirmed with a computed tomography (CT) scan;Arterial partial pressure of oxygen (PaO_2_) < 60 mmHg when breathing ambient air;Pulmonary hypertension (PH) ≥ 40 mmHg based on Doppler ultrasonography;Informed consent of the patient or their legal representative to participate in the study.

Exclusion criteria:Circulatory instability with 80 mmHg < systolic blood pressure < 180 mmHg;High risk of pulmonary embolism (PE) as defined by ESC Guidelines [22];Clinically significant active bleeding;Kidney insufficiency with creatinine clearance (Cockcroft-Gault formula) < 30 mL/min;Severe liver failure;Retinitis pigmentosa;Hypersensitivity to sildenafil or enoxaparin sodium;Pregnancy or breastfeeding;Participation in other clinical trials.

All patients participating in the study received the standard treatment for severe COVID-19 according to the local protocol:methylprednisolone 0.5 mg/kg daily orally;enoxaparin 60 anti-Xa IU/kg once daily if D-dimer < 5 µg/mL, or 100 anti-Xa IU/kg once daily if D-dimer levels ≥ 5 µg/mL;antibiotics in the presence of pathogenic bacterial flora in sputum, urine, or blood;an infusion volume calculated to ensure zero or slightly negative daily fluid balance;oxygen therapy with or without non-invasive ventilation to maintain respiratory index PaO_2_/FiO_2_ > 100 mmHg;intubation and pressure-controlled mechanical ventilation if PaO_2_/FiO_2_ < 100 mmHg;standard treatment of comorbidities.

Patients were randomly assigned to one of two groups:

Group 1 (treatment group, S)—sildenafil (Superviga, Zdorovye, Ukraine) was administered orally at a dose of 0.25 mg/kg three times a day for seven days after enrolment;

Group 2 (control group, C)—placebo was administered in a double-blind manner.

The following parameters were measured in all patients one day before the start of the treatment and seven days after the start of the treatment: complete blood count, arterial blood gases, C-reactive protein (CRP), procalcitonin (PCT), fibrinogen, D-dimer, Il-6, and ferritin. Doppler measurement of pulmonary artery pressure (PAP) was also performed.

### 2.3. Statistical Analysis

MedCalc^®^ Statistical Software version 20.218 (MedCalc Software Ltd., Ostend, Belgium; https://www.medcalc.org; 2023, accessed on 20 January 2023) was used in the analysis. The data distribution in the examined population was different from normal, as demonstrated by the Shapiro–Wilk test. Thus, the median (Me) and interquartile range (QI–QIII) were calculated for quantitative data and frequency (%) for qualitative data. The Mann–Whitney test was used to compare quantitative variables. Fisher’s exact test was used to compare the value of qualitative features between the two groups. Logistic regression models were used to estimate the impact of potential risk factors on intubation and death. The inherent ability of the models to discriminate between the subjects who met or failed to meet the predefined endpoints (intubation or death) was assessed using the receiver operating characteristic (ROC) method. The Youden index (sensitivity + specificity − 1) was used to find the optimal threshold (optimal Y value) for the tested dependencies. The area under the ROC curve (AUC) and its 95% confidence intervals (CI) were also calculated. The odds ratio (OR) for independent variables and 95% CI were calculated to estimate the relative amount by which the odds of the outcome increased (OR > 1) or decreased (OR < 1) when the value of the independent variable was increased by 1 unit. The *p*-value < 0.05 was considered significant for all statistical tests.

## 3. Results

The demographic, clinical, and laboratory characteristics of the study groups of 75 participants each are shown in Table 1. The median values of the investigated parameters did not differ significantly between the groups, with the exception of age.

In the sildenafil group, 19 patients (25.3%) did not receive a COVID-19 vaccine, 37 (49.3%) were vaccinated with one dose, and 19 (25.3%) with two doses. In the control group, 18 patients (24%) were not vaccinated at all, 39 (52%) received one, and 18 (24%) received two doses. There were no statistically significant differences between groups (*p* = 0.948). 

Figure 1 presents the mean sPAP values before and after treatment (∆sPAP) with sildenafil and the placebo, respectively. Sildenafil treatment resulted in a sPAP decrease with ∆sPAP = 11 mm Hg, i.e., from 63 mm Hg to 52 mm Hg (*p* < 0.001; 95% CI 10.0–12.0), while, for the placebo, ∆sPAP was only 1 mm Hg (*p* = 0.014; 95% CI 0.0–1.46).

During the 7-day treatment phase, the mortality rates were 4% and 13.3% (*p* = 0.078 using Fisher’s exact test), and the intubation rates were 8% and 18.7% (*p* = 0.091 using Fisher’s exact test), for groups S and C, respectively.

Logistic regression was used to identify variables associated with the risk of death or intubation. Table 2 shows the results of a single-factor logistic regression analysis. The single-factor analysis demonstrated that the risk of death was correlated with increased sPAP levels and D-dimer and ferritin concentrations, as well as a decreased respiratory index and thrombocyte count. The impact of sildenafil treatment was borderline statistically significant, with *p* = 0.055.

Next, using multivariate logistic regression models, which of these variables formed a set of factors related to the risk of death was examined. A stepwise approach was used (entering a variable if *p* < 0.1 and removing a variable if *p* > 0.2). Two significant risk factors were identified: sPAP and treatment with sildenafil. Table 3 shows the results of the coefficient analysis of the model.

Multivariate analysis demonstrated that the risk of death increased with increasing sPAP values and decreased with treatment S (when standardized to the initial sPAP). Figure 2 presents the ROC curve predicting the risk of death in the two-factor model.

The area under the ROC curve, AUC = 0.86 (95% CI 0.80–0.91), indicates a strong correlation between the risk of death and both sPAP and the treatment. The optimal threshold (using the Youden index) for death prediction Y_crit_ was higher than 0.0879, with sensitivity of 92.3% (95% CI 64.0%–99.8%) and specificity of 75.2% (95% CI 67.1%–82.2%). It should be noted that the model predicted death at sPAP > 63 mmHg and > 67 mmHg for the placebo and sildenafil groups, respectively.

Logistic regression models were used to identify factors associated with the risk of intubation (Table 4). The same variables were considered potentially significant as for the risk of death.

Again, to identify a set of variables associated with the risk of intubation, multivariate logistic regression analysis using a stepwise approach (entering a variable if *p* < 0.1, removing a variable if *p* > 0.2) was used. Two risk factors were identified: initial sPAP and treatment. Table 5 shows the values of the model coefficients.

Multivariate analysis revealed that the risk of intubation increased with increasing sPAP values and decreased in patients treated with sildenafil compared to the those receiving a placebo (if standardized to initial sPAP). Figure 3 shows the ROC curve predicting the intubation risk in the two-factor model.

The AUC = 0.86 (95% CI 0.80–0.91) indicates a strong association between the risk of intubation and treatment with the placebo. The model used predicted intubation at sPAP > 63 mmHg and >66 mmHg for the placebo and sildenafil groups, respectively.

In patients who survived, the median ICU stay was 15 (11–26) days, and it was 19 (13–29) days for the sildenafil and control groups, respectively. The difference was statistically significant with *p* < 0.001; the data are shown in Figure 4.

In patients with severe COVID-19 and pulmonary hypertension, treatment with sildenafil resulted in a five-times lower risk of death, a four-times lower risk of intubation with invasive mechanical ventilation, and a shorter ICU hospitalization period by 4 days compared to treatment with the placebo.

## 4. Discussion

Our research demonstrated that in patients with severe COVID-19 and pulmonary artery hypertension, treatment with sildenafil resulted in a five times lower risk of death, four times lower risk of intubation with invasive mechanical ventilation, and a shorter ICU hospitalization period by 4 days compared to treatment with the placebo. This is only if adjusted for PAH, and, without this adjustment, the differences in mortality and intubation rates did not demonstrate statistical significance.

The main mechanism of sildenafil’s action is vasodilatation due to PDE-5 inhibition. In our study, after a 7-day treatment period, an 11 mm Hg (from 63 to 52 mm Hg, 17.5%, *p* < 0.001) decrease in the average sPAP was statistically significant in patients treated with sildenafil, as compared to only a 1 mm Hg decrease in the placebo group (Figure 1). A meta-analysis by Wu et al. of over 200 patients with pulmonary hypertension secondary to chronic systolic heart failure reported PDE-5 inhibitors as an add-on therapy to reduce the mean PAP by 5.71 mm Hg (*p* < 0.05) [23]. Another meta-analysis comprising 928 similar cardiac patients demonstrated an insignificant decrease in the mean PAP, while sPAP was significantly reduced by 11.52 mm Hg (95% CI −15.56, −7.49; *p* < 0.001) [24]. Arif et al. showed, in patients with COPD-related PAH, that PDE-5 inhibitors significantly decreased sPAP (pooled treatment effect 5.9 mm Hg; 95% CI −10.3, −1.6, *p* = 0.007), but had inconsistent clinical benefits [25]. A meta-analysis by Barnes et al. including 174 patients with all-cause PAH treated with sildenafil demonstrated a pooled decrease in PAP by 7.34 mm Hg (95% CI −9.35, −5.33) [26]. A study exploring the effect of sildenafil on sPAP demonstrated an insignificant decrease from 105.23 +/− 17.82 mm Hg to 98.50 +/− 24.38 mm Hg in 22 patients [27]. Thus, we can consider sildenafil to be equally effective at reducing PAH in COVID-19 patients as in patients with PAH due to other causes.

In addition to its vasodilative effect, sildenafil was proven to have several other properties [28,29]. Sildenafil treatment seems to be substantiated by its anti-inflammatory, antioxidant, immunomodulatory, and anti-apoptotic activity. Consequently, it may have an impact on activated T-cell modulation, reduced cytokine release, increased oxygen diffusion, and the stimulation of vascular recovery, the key factors in the pathomechanism of COVID-19.

Data about sildenafil administration in severe COVID-19 patients are surprisingly rare and the groups studied are very small. Santamarina et al. [20] recently published the results of a randomized, pilot study focused on the sildenafil effect on oxygenation parameters in patients with severe COVID-19 and respiratory insufficiency. The study included 20 participants each in the sildenafil and control groups, with baseline echocardiography suggesting PAH, right ventricular dysfunction (RVD), or both. McFadyen et al. [21] published a retrospective study on 25 critically ill COVID-19 patients (10 of them on ECMO—extracorporeal membrane oxygenation), again with PAH, RVD, or both, all treated with sildenafil. The study examined the impact of sildenafil on gas exchange. Ouamar et al. [19] described their experience with the long-term administration of sildenafil to patients with RVD or PAH secondary to acute pulmonary thrombosis in the course of severe COVID-19. Twenty-three consecutive patients were enrolled and again no controls were used.

The primary endpoint in our study was mortality. The mortality rate was 4% and 13.3% (*p* = 0.078) in the sildenafil and control groups, respectively. Santamarina et al. reported no deaths in the treatment group and 5% among the control group [20]. In our study, the risk of death was influenced by sPAP, D-dimer, ferritin, the respiratory index, and thrombocytes; all these features are considered hallmarks of COVID-19 severity. In this single-factor analysis, treatment with sildenafil with *p* = 0.055 presented borderline significance (Table 2). However, if adjusted for sPAP, treatment with sildenafil reduced the risk of death almost fivefold (Table 3). Very similar results in all-cause PAH were presented by Barnes et al. In five RCTs with 442 participants on sildenafil and 192 controls, they found significantly reduced mortality (OR 0.23, 95% CI 0.05–0.98) in the treatment group compared to the control group [26].

Almost identical results we obtained for intubation. Again, the difference in the intubation rate was statistically insignificant between the groups (8% vs. 18.7%; *p* = 0.09) and the impact of sildenafil treatment on the intubation rate in a single-factor analysis was also insignificant (Table 4). In addition, as in the case of mortality, adjustment for sPAP yielded almost a fourfold decrease in the risk of intubation (Table 5). A beneficial effect of sildenafil on intubation was also documented by Santamarina et al., who reported no intubation vs. a 20% intubation rate (*p* = 0.04) in the treatment and placebo groups, respectively [20].

The length of ICU hospitalization was the third parameter investigated in our study. Sildenafil treatment resulted in a statistically significant shortening in mean ICU stays from 19 days in the placebo group to 15 days in the treatment group (Figure 4). An even larger but insignificant difference of 15 vs. 7 days was noted by Santamarina et al. [20].

Overall, our results agreed with the infrequent data from other researchers: sildenafil treatment yields beneficial results, but not in every case, and sometimes statistical significance was achieved. We demonstrated that the clinical effectiveness of this treatment is positive and significant but only if adjusted for PAH. This allows us to speculate that among COVID-19 patients, the higher the PAH, the more effective sildenafil treatment may be. Consequently, these patients should be considered a target group for treatment with sildenafil.

Our study has some limitations. Firstly, although three centers were involved, they covered a population limited by territory, with only 75 participants in each arm. Secondly, the treatment only lasted one week, with no follow-up period. A statistically significant clinical effect could probably be achieved if the duration of treatment and follow-up were longer. Thus, it seems reasonable to verify the results obtained in a multicenter study with more participants and longer treatment and follow-up periods, and perhaps with a more selective group of patients.

## 5. Conclusions

Treatment with sildenafil seems to reduce the risk of death, risk of intubation, and duration of ICU hospitalization in patients with severe COVID-19 and pulmonary arterial hypertension (PAH). Sildenafil should be considered as an add-on therapy in these patients. Further studies are necessary to confirm the usefulness of sildenafil in patients with severe COVID-19 and pulmonary arterial hypertension.

## Figures and Tables

**Figure 1 viruses-15-01157-f001:**
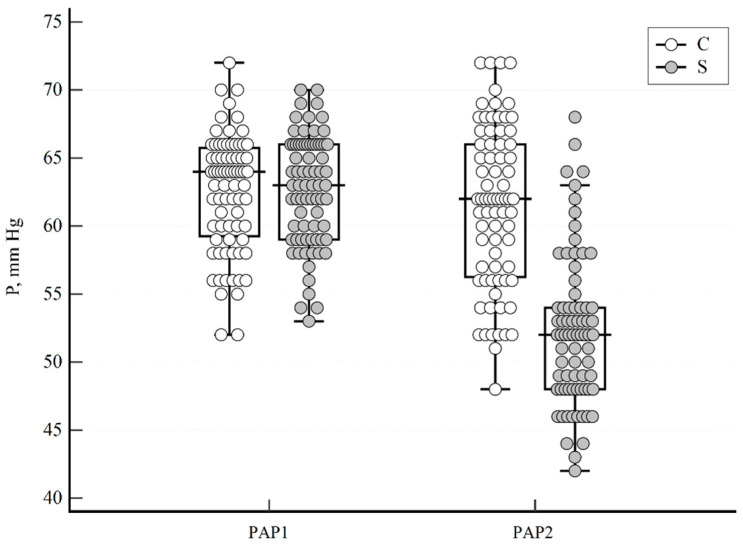
Systolic pulmonary artery pressure before (sPAP1) and after (sPAP2) treatment with sildenafil (S) or the placebo (C), respectively. Solid circles—treatment with sildenafil; empty circles—treatment with placebo. Median, minimum, and maximum values and interquartile ranges are presented.

**Figure 2 viruses-15-01157-f002:**
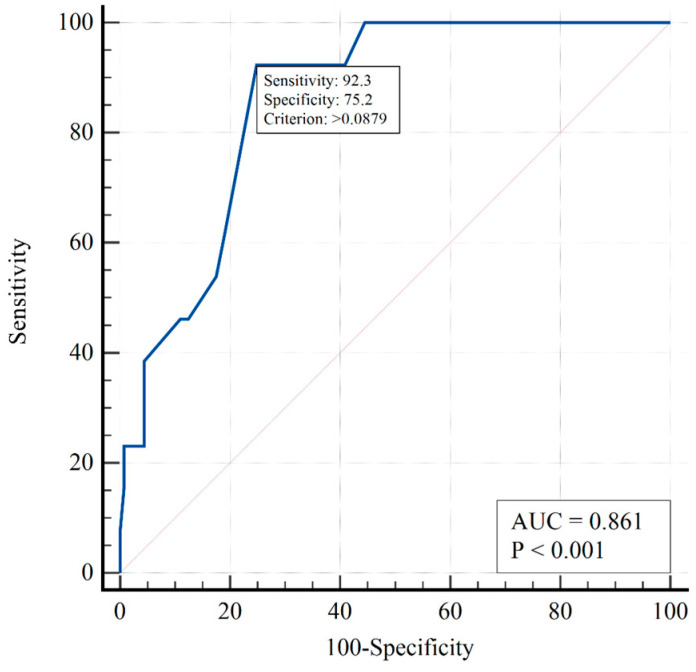
ROC curve predicting risk of death in the two-factor logistic regression model.

**Figure 3 viruses-15-01157-f003:**
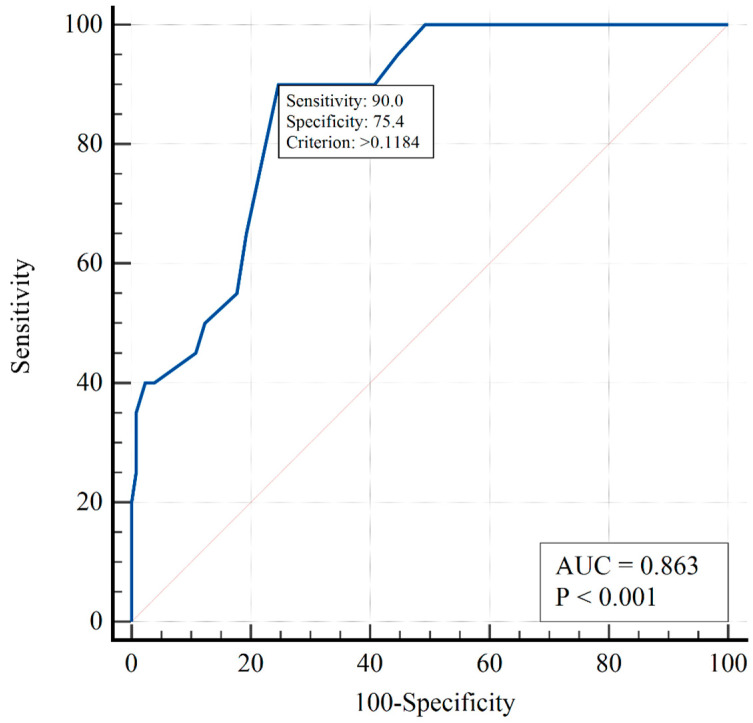
ROC curve predicting risk of intubation in the two-factor logistic regression model.

**Figure 4 viruses-15-01157-f004:**
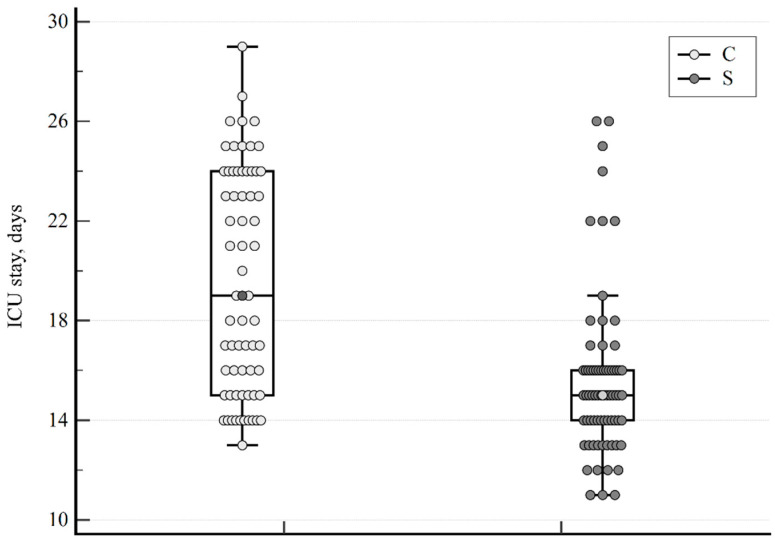
Duration of ICU hospitalization for surviving patients treated with sildenafil (S) or placebo (C), respectively. Median, minimum, and maximum values and interquartile ranges are presented.

**Table 1 viruses-15-01157-t001:** Basic clinical and laboratory data in the study groups, median (QI–QIII) for continuous variables, and numerical values (%) for categorical variables.

	S (*n* = 75)	C (*n* = 75)	*p* Value
Age, years	70 (68–72)	68 (66–71)	0.012
Sex, female, %	46.7	46.7	1.0
Interleukin-6, pg/mL	60 (34–72)	68 (49–76)	0.256
Procalcitonin, ng/mL	0.2 (0.1–0.375)	0.2 (0.2–0.3)	0.991
Fibrinogen, g/L	6.5 (5.7–7.5)	7.5 (6.5–7.5)	0.041
CRP, mg/L	52 (39–72)	56 (48–72)	0.834
D-dimer, µg/L	1246 (439–1652)	1340 (439–1657)	0.532
Leukocytes, ×10^9^/L	4.2 (3.5–4.3)	4.2 (4–4.3)	0.543
Thrombocytes, ×10^9^/L	126 (101.25–138)	126 (87–135.5)	0.59
Lymphocytes, %	24 (22–26)	24 (22–26)	0.47
Erythrocytes, ×10^12^/L	3.2 (2.6–3.675)	3.2 (2.6–3.675)	0.809
PaO_2_/FiO_2_, mm Hg	116 (111–124)	118 (112–129)	0.083
Ferritin, ng/mL	998 (597.5–1319.25)	908 (661–1335)	0.744
sPAP, mm Hg	63 (59–66)	64 (59.25–65.750)	0.773

sPAP—systolic pulmonary artery pressure; C—control group; S—sildenafil group. Laboratory test reference ranges: interleukin-6 < 4.0 pg/mL, procalcitonin < 0.02 ng/mL, fibrinogen 2.0–4.0 g/L, CRP < 5.0 mg/L, D-dimer < 500 µg/L, leukocytes 4.0–9.0 × 10^9^/L, lymphocytes 19–37%, thrombocytes 200–400 × 10^9^/L, erythrocytes 3.6–4.2 × 10^12^/L, ferritin 8–143 ng/mL, PaO_2_/FiO_2_ 454–495 mm Hg, sPAP 20–35 mm Hg.

**Table 2 viruses-15-01157-t002:** Analysis of risk of death in a single-factor logistic regression model.

Independent Variables		Model Coefficient, b ± m	Significance Level of Difference in the Coefficient from 0, *p* Value	Odds Ratio, OR (95% CI)
Treatment	C	Reference		
	S	−1.31 ± 0.68	0.055	0.27 (0.07–1.03)
Sex	female	Reference		
	male	0.02 ± 0.58	0.969	–
Age, per year		−0.084 ± 0.062	0.180	–
Interleukin-6, pg/mL		−0.007 ± 0.015	0.638	−
Procalcitonin, ng/mL		−2.25 ± 2.46	0.360	−
Fibrinogen, g/L		−0.27 ± 0.36	0.457	−
CRP, mg/L		0.003 ± 0.017	0.864	−
D-dimer, per mg/L		0.24 ± 0.05	<0.001	1.27 (1.15–1.42)
Leukocytes, ×10^9^/L		0.21 ± 0.19	0.268	–
Lymphocytes, %		0.041 ± 0.086	0.638	–
Thrombocytes, ×10^9^/L		−0.033 ± 0.013	0.012	0.97 (0.94–0.99)
Erythrocytes, ×10^12^/L		0.44 ± 0.49	0.357	–
PaO_2_/FiO_2_, per 10 mm Hg		−0.87 ± 0.36	0.015	0.42 (0.21–0.84)
Ferritin, per mg/L		0.29 ± 0.08	0.003	1.33 (1.14–1.56)
sPAP, mm Hg		0.42 ± 0.12	0.001	1.53 (1.20–1.94)

sPAP—systolic pulmonary artery pressure; C—control group; S—sildenafil group. Laboratory test reference ranges: interleukin-6 < 4.0 pg/mL, procalcitonin < 0.02 ng/mL, fibrinogen 2.0–4.0 g/L, CRP < 5.0 mg/L, D-dimer < 500 µg/L, leukocytes 4.0–9.0 × 10^9^/L, lymphocytes 19–37%, thrombocytes 200–400 × 10^9^/L, erythrocytes 3.6–4.2 × 10^12^/L, ferritin 8–143 ng/mL, PaO_2_/FiO_2_ 454–495 mm Hg, sPAP 20–35 mm Hg.

**Table 3 viruses-15-01157-t003:** Multivariate logistic regression model for predicting the risk of mortality.

Independent Variables		Model Coefficient, b ± m	Significance Level of Difference in the Coefficient from 0, *p* Value	Odds Ratio, OR (95% CI)
Treatment	C	Reference		
	S	–1.56 ± 0.74	0.035	0.21 (0.05–0.89)
sPAP		0.46 ± 0.13	0.001	1.58 (1.21–2.05)

sPAP—systolic pulmonary artery pressure; C—control group; S—sildenafil group.

**Table 4 viruses-15-01157-t004:** Single-factor logistic regression model for predicting risk of intubation.

Independent Variables		Model Coefficient, b ± m	Significance Level of Difference in the Coefficient from 0, *p* Value	Odds Ratio, OR (95% CI)
Treatment	C	Reference		
	S	−0.97 ± 0.52	0.061	0.38 (0.14–1.05)
Sex	female	Reference		
	male	0.08 ± 0.48	0.873	–
Age, per year		–0.050 ± 0.052	0.331	–
Interleukin-6, pg/mL		–0.005 ± 0.012	0.718	–
Procalcitonin, ng/mL		–0.46 ±1.95	0.814	–
Fibrinogen, g/L		–0.27 ± 0.30	0.367	–
CRP, mg/L		0.015 ± 0.014	0.287	–
D-dimer, per mg/L		0.17 ± 0.04	<0.001	1.19 (1.10–1.28)
Leukocytes, ×10^9^/L		0.35 ± 0.15	0.020	1.42 (1.06–1.91)
Lymphocytes, %		−0.054 ± 0.066	0.411	
Thrombocytes, ×10^9^/L		−0.026 ± 0.010	0.011	0.97 (0.96–0.99)
Erythrocytes, ×10^12^/L		0.27 ± 0.39	0.486	–
PaO_2_/FiO_2_, per 10 mm Hg		−1.42 ± 0.39	<0.001	0.42 (0.21–0.84)
Ferritin, per mg/L		0.22 ± 0.06	<0.001	1.25 (1.10–1.41)
sPAP, mm Hg		0.49 ± 0.12	<0.001	1.63 (1.29–2.05)

sPAP—systolic pulmonary artery pressure; C—control group; S—sildenafil group. Laboratory test reference ranges: interleukin-6 < 4.0 pg/mL, procalcitonin < 0.02 ng/mL, fibrinogen 2.0–4.0 g/L, CRP < 5.0 mg/L, D-dimer < 500 µg/L, leukocytes 4.0–9.0 × 10^9^/L, lymphocytes 19–37%, thrombocytes 200–400 × 10^9^/L, erythrocytes 3.6–4.2 × 10^12^/L, ferritin 8–143 ng/mL, PaO_2_/FiO_2_ 454–495 mm Hg, sPAP 20–35 mm Hg.

**Table 5 viruses-15-01157-t005:** Multivariate logistic regression model for predicting the risk of intubation.

Independent Variables		Model Coefficient, b ± m	Significance Level of Difference in the Coefficient from 0, *p* Value	Odds Ratio, OR (95% CI)
Treatment	C	Reference		
	S	−1.356 ± 0.61	0.027	0.26 (0.08–0.86)
sPAP		0.53 ± 0.13	<0.001	1.70 (1.32–2.20)

sPAP—systolic pulmonary artery pressure; C—control group; S—sildenafil group.

## Data Availability

The data presented in this study are available on request from the corresponding author. The data are not publicly available due to ethical aspects.

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
