# Peer review of "Efficacy of Sildenafil in Patients with Severe COVID-19 and Pulmonary Arterial Hypertension"

_viruses, 2023, doi:10.3390/v15051157_

Round 1

Reviewer 1 Report

Thanks for the invitation. This study focuses on one of the ongoing hot topics, but some modifications need to be made. 

Other comments are:

  1. the sample size of 150 is too small to get conclusive results
  2. the No. of the study participants should be added to the results section
  3. I couldn't find the gap of knowledge ( what will your study add to the previous studies)
  4. Did authors noticed difference among gender in drug efficacy?
  5. the bullets should be removed from the conclusion section
  6. Last, if you agree with the following suggestion, I would like to add a timeline image to show the entire process of research

Author Response

Dear Reviewer, thank you very much for the review and valuable comments.

Please find the answers to your questions below:

  1. the sample size of 150 is too small to get conclusive results

Yes, we agree that the sample size is too small for conclusive results - this was pointed out as the first limitation of the study.

  1. the No. of the study participants should be added to the results section

Added, line 148

  1. I couldn't find the gap of knowledge (what will your study add to the previous studies)

As mentioned in the discussion, there are only few studies on the use of sildenafil in severe COVID-19 and all of them were conducted on very small samples. Our study covered a substantially larger sample and control group. Given limited number of studies, each subsequent study on another population, with different inclusion/ exclusion criteria, is highly additive, even if the results are more or less consistent with those of other researchers. Confirmation of certain correlations allows to chart new treatment paths.

  1. Did authors noticed difference among gender in drug efficacy?

There were no differences between groups with respect to the number of women/men. We did not split the studied groups S and C regarding gender, because the groups would be too small and conclusions much weaker.

  1. the bullets should be removed from the conclusion section.

Removed.

  1. Last, if you agree with the following suggestion, I would like to add a timeline image to show the entire process of research

In the end, we decided not to include a timeline image, because the work contains many figures and tables and the study itself is relatively complex. Needless to say, should this be a condition of acceptance, we will prepare it.

Reviewer 2 Report

In the current manuscript, the authors describe the results of a clinical study designed to evaluate the clinical benefit(s) of sildenafil administration in patients diagnosed with severe COVID-19 and pulmonary arterial hypertension (PAH). The authors' study design (setting, criteria, measured outcomes) and selected methods (i.e., parameters measured) are appropriate and scientifically sound. Though not all of the study's findings achieved statistically significance, the observed trends in reduced mortality rates, reduced risk for intubation, and reduction in length of ICU stay are highly noteworthy. Further, these results support larger-scale clinical trials to assess the clinical effectiveness of sildenafil treatment in PAH patients, as well as other high risk groups. My comments: 

1. The inclusion of additional key demographic data would be of interest to informed readers and help guide the design of future studies. Such data includes: racial/ethnic make-up of both groups, average time between symptom onset and hospitalization, and whether participants are primarily free-living or community-dwelling.

2. Considering the 2-year time period of the study and the ongoing waves of viral variants, it is important for the authors to comment on whether any differences were observed in their measured outcomes between individuals admitted early in the study to those admitted later.

3. Likewise, it is important for the authors to comment on the vaccination status of participants in both arms (i.e., % of individuals receiving primary series, % of individuals receiving booster doses).

Author Response

Dear Reviewer, thank you very much for the review and valuable comments.

Please find the answers to your questions below:

  1. The inclusion of additional key demographic data would be of interest to informed readers and help guide the design of future studies. Such data includes: racial/ethnic make-up of both groups, average time between symptom onset and hospitalization, and whether participants are primarily free-living or community-dwelling.

We have added racial information (lines: 76-77). Unfortunately, we did not analyze the duration of symptoms or assess social status.

  1. Considering the 2-year time period of the study and the ongoing waves of viral variants, it is important for the authors to comment on whether any differences were observed in their measured outcomes between individuals admitted early in the study to those admitted later.

We noted no differences as to therapy efficacy between viral variants. Given the duration of the study and the varying number of severe cases in each wave, and a notable decrease during the omicron wave, we decided not to publish such data.

  1. Likewise, it is important for the authors to comment on the vaccination status of participants in both arms (i.e., % of individuals receiving primary series, % of individuals receiving booster doses).

We added this information (lines: 159-162). As the patients analyzed were hospitalized between 2020 and 2022 (different virus variants) and due to quite small studied groups we are unable to draw any reliable conclusions.